# *Oncogenic NRAS* Accelerates Rhabdomyosarcoma Formation When Occurring within a Specific Time Frame during Tumor Development in Mice

**DOI:** 10.3390/ijms222413377

**Published:** 2021-12-13

**Authors:** Nada Ragab, Julia Bauer, Dominik S. Botermann, Anja Uhmann, Heidi Hahn

**Affiliations:** Department of Human Genetics, University Medical Center Goettingen, 37073 Goettingen, Germany; nada.ragab@med.uni-goettingen.de (N.R.); julia.bauer89@gmx.de (J.B.); dominik.botermann@med.uni-goettingen.de (D.S.B.); auhmann@gwdg.de (A.U.)

**Keywords:** RAS mutation, embryonal RMS, permissive window

## Abstract

In the *Ptch+/-* mouse model for embryonal rhabdomyosarcoma (ERMS), we recently showed that oncogenic (onc) H-, K- or NRAS mutations do not influence tumor growth when induced at the advanced, full-blown tumor stage. However, when induced at the invisible ERMS precursor stage at 4 weeks of age, tumor development was enforced upon oncHRAS and oncKRAS but not by oncNRAS, which instead initiated tumor differentiation. These data indicate that oncRAS-associated processes differ from each other in dependency on the isoform and their occurrence during tumor development. Here, we investigated the outcome of oncNRAS induction at an earlier ERMS precursor stage at 2 weeks of age. In this setting, oncNRAS accelerates tumor growth because it significantly shortens the ERMS-free survival and increases the ERMS incidence. However, it does not seem to alter the differentiation of the tumors. It is also not involved in tumor initiation. Together, these data show that oncNRAS mutations can accelerate tumor growth when targeting immature ERMS precursors within a specific time window, in which the precursors are permissive to the mutation and show that oncNRAS-associated processes differ from each other in dependency on their occurrence during tumor development.

## 1. Introduction

Rhabdomyosarcoma (RMS) is the most common soft tissue sarcoma in children and comprises 4.5% of all childhood cancer with an annual incidence of 4.5 cases per million children. Although 90% of patients with a low-risk tumor can be cured by multi-modal therapies, the overall survival rates of patients with metastatic or recurrent disease are very low at 21% and 30%, respectively [1].

Based on histopathologic features, RMS is classified into embryonal RMS (ERMS), alveolar RMS (ARMS), pleomorphic, spindle cell and sclerosing tumors [1,2]. The two primary RMS subtypes in children are ARMS and ERMS. ERMS represents the majority of cases and is associated with a more favorable prognosis than ARMS. Relevant in etiology and the poorer prognosis of ARMS are chromosomal translocations resulting in PAX3-FOXO1 or PAX7-FOXO1 fusion proteins that occur in approximately 80% of ARMS [3]. The fraction of fusion-negative ARMS is molecularly and clinically similar to ERMS [4]. 

Recently, 641 RMS samples from patients enrolled on Children’s Oncology Group trials (1998–2017) and UK patients enrolled on malignant mesenchymal tumor and RMS2005 trials (1995–2016) were subjected to sequencing. Whereas the most frequent lesions in fusion-positive tumors were amplification of the *Cyclin-Dependent Kinase 4* (*CDK4*) or the *MYCN* gene, the most frequently observed alteration in fusion-negative tumors was oncogenic RAS (oncRAS) isoform mutations that occurred in 32% of cases. Whereas *KRAS* and *HRAS* mutations were found in 9% and 8% of fusion-negative tumors, respectively, *NRAS* mutations were observed in 17% of cases [5]. This demonstrates that oncNRAS is the most frequent mutation in ERMS, which also has been shown in other studies [6,7,8].

In cancer, oncogenic single-point mutations of all RAS isoforms most commonly affect codons 12, 13 and 61, and all of them impair GTP hydrolysis. This renders RAS preferentially GTP-bound and active [9,10]. Interestingly, the oncRAS isoform as well as the position and the substitution of the mutations vary between human tumor subtypes [10]. 

We recently compared the effects of oncRAS mutations affecting codon 12 of all 3 RAS isoforms (i.e., oncK-, oncH- and oncNRAS) in the *Ptch+/-* mouse model for ERMS-like tumors. In this model, ERMS are initiated before birth and become palpable at the earliest around 7 weeks of age [11]. Our data show that none of the oncRAS isoforms affect ERMS growth when induced at the full-blown, palpable tumor stage. In contrast, when induced at the tumor precursor stage at 4 weeks of age, oncKRAS and oncHRAS mutations reinforce tumor development, whereas oncNRAS does not and rather induces a more differentiated phenotype [12]. 

Here, we induced the oncNRAS mutation at an earlier stage during tumor formation in 2-week-old mice and analyzed the impact on incidence, growth, latency time, multiplicity and differentiation status of the tumors.

## 2. Results and Discussion

As described above, we recently generated *Ptch^+/-^* mice that harbor a conditional oncogenic *NRasG12D* (*NRas^fl^*) allele [12]. To activate the expression of oncNRAS in ERMS of *Ptch^+/-^**NRas^fl/+^* mice, the animals were bred to *Myf5^CreER^* mice that express a tamoxifen-inducible Cre recombinase in *Myf5*-expressing cells [12,13]. Using this model, we showed that induction of oncNRAS in 4-week-old *Ptch^+/-^* mice does not influence incidence, growth or proliferation of the tumors when compared to the controls [12] (summarized in Table 1). We now induced the oncNRAS mutation at an earlier tumor stage. For this purpose, we again crossed *Ptch^+/-^**NRas^fl/+^* to *Myf5^CreER/CreER^* mice. Thus, all offspring were heterozygous for *Myf5^CreER^* and either *Ptch^+/-^**NRas^fl/+^, Ptch^+/+^**NRas^fl/+^, Ptch^+/-^NRas^+/+^* or *Ptch^+/+^**NRas^+/+^* (in the following the term “*Myf5^CreER/+^*” is omitted when describing the respective mice). All pups were treated twice with tamoxifen at postnatal days 12 and 14 and were genotyped upon weaning (Table 1).

Mendelian inheritance was not affected, even if the number of born mice with the *NRas^+/+^* genotype seemed to be slightly reduced compared to the number of born *NRas^fl/+^* siblings (27 *Ptch^+/-^**NRas^fl/+^* vs. 16 *Ptch^+/-^**NRas^+/+^* and 28 *Ptch^+/+^**NRas^fl/+^* vs. 20 *Ptch^+/+^**NRas^+/+^* mice, *p* = 0.1884 by Chi-Quadrat testing, Table 1). A deviation in gender distribution was also not observed (data not shown). 

Next, the tumor incidence of the animals was examined. For this purpose, ERMS development was monitored weekly by manual palpation for about 200 days, if possible (Table 1). At the end of monitoring, sacrificed mice were thoroughly examined for additional non-palpable tumors. Tumors were carefully dissected, and ERMS were identified on H&E-stained paraffin sections. ERMS with negligible amounts of fat, necrosis or normal skeletal muscle were subjected to molecular analyses.

ERMS occurred exclusively in mice with the *Ptch^+/-^* genotype, and no ERMS was found in *Ptch^+/+^**NRas^+/+^* and *Ptch^+/+^**NRas^fl/+^* mice (Table 1; one *Ptch^+/+^**NRas^fl/+^* offspring had an opacity of the eye lens). Remarkably, oncNRAS significantly reduced the latency time until detection of palpable ERMS (median latency 66 days or 108 days for *Ptch^+/-^**NRas^fl/+^* or *Ptch^+/-^**NRas^+/+^* mice, respectively) and shortened ERMS-free survival of the animals (Figure 1a, *p* = 0.0473 by Wilcoxon testing). The lifespan between *Ptch^+/-^**NRas^fl/+^* and *Ptch^+/-^**NRas^+/+^* mice was not significantly different (Figure 1b, *p* = 0.0944 by Wilcoxon testing), even if too large tumors or a bad general condition required premature termination and contributed to a reduced overall survival of the mice in a few cases. In addition, the ERMS incidence at the end of the study was significantly higher in *Ptch^+/-^**Nras^fl/+^* mice (Table 1; Figure 1c top, *p* = 0.0298 by Chi-Quadrat testing). This was not only due to the occurrence of palpable ERMS but also of non-palpable ERMS, which was never observed in *Ptch^+/-^**Nras^+/+^* mice (Table 1; Figure 1c middle, *p* = 0.0468 by Chi-Quadrat testing). On the other hand, oncNRAS did not significantly influence tumor multiplicity (i.e., mice with ≥2 ERMS; Table 1 and Figure 1c bottom, *p* = 0.0925 by Chi-Quadrat testing) and also did not influence weight (Figure 1d, left and middle) or the proliferation rate of the tumors (Figure 1d, right). 

The efficient recombination of the floxed *NRas* locus in *NRas^fl/+^* mice after tamoxifen application was verified in all mice carrying the *NRas^fl^* allele. The recombination of the floxed *NRas* locus was not limited to ERMS but was also observed in normal skeletal muscle of *Ptch^+/-^**NRas^fl/+^* mice (one example of recombination in skeletal muscle and ERMS of *Ptch^+/-^**NRas^fl/+^* mice in comparison to the *Ptch^+/-^**NRas^+/+^* controls is shown in Figure 2a left). Indeed, the floxed *NRas* locus was consistently recombined in all mice carrying the *NRasfl*/+ allele (recombination in skeletal muscle of *Ptch^+/+^**NRas^fl/+^* mice compared to *Ptch^+/-^**NRas^fl/+^* mice is shown in Figure 2a right). The recombination of the floxed *NRas* allele increased RAS activity not only in ERMS (Figure 2b upper panel) but also in normal muscle tissue of *Ptch^+/+^**NRas^fl/+^* mice (Figure 2b bottom), which never developed ERMS. Therefore, a single oncNRAS mutation in normal muscle tissue of 2-week-old wild-type mice apparently does not lead to ERMS formation.

In summary, our data show several things. First, the data suggest that oncNRAS expression in 2-week-old mice is involved in the acceleration of tumor formation but not in tumor initiation. This is due to the fact that oncNRAS per se does not result ERMS formation but increases ERMS incidence in ERMS-prone *Ptch^+/-^* mice without affecting the tumor load.

Secondly, the data implicate that the oncNRAS mutation must occur within a very specific time window during tumor development to make the tumor more aggressive and that this window closes between the age of 2 and 4 weeks, at least in *Ptch^+/-^* mice. This is due to the fact that oncNRAS accelerates ERMS formation when expressed in 2-week-old *Ptch^+/-^* mice but has no impact on ERMS formation when the animals are 4 weeks old [12]. In addition, this window must open after fertilization because an oncogenic NRASG12D germline mutation is lethal and leads to the death of the E15.5 to E17.5-old mouse embryos [14,15]. This might be similar in humans with Noonan syndrome, which is caused by germline mutations in NRAS and which predisposes the patients to ERMS development. Thus, although mutations at the NRAS G12 codon have been discovered in patients with this syndrome, ERMS-prototypic oncNRAS mutations, such as the G12D mutation or Q61K mutations [10], have never been found [16,17,18].

Finally, we investigated whether the expression of oncNRAS modulates the expression of muscle differentiation markers. To this end, we have investigated the expression of myogenin (MYOG), which controls the terminal differentiation of myoblasts into myocytes [19] and plays an important role in the differentiation of RMS [20]. We also examined the expression of desmin (DES), which is a myoblast marker [21], and of the late muscle differentiation marker tropomyosin 3 (TPM3). As seen in Figure 3, ERMS from *Ptch^+/-^**NRas^+/+^* and *Ptch^+/-^**NRas^fl/+^* mice were histologically identical (Figure 3a), and the protein expression of all three markers did not significantly differ between *Ptch^+/-^**NRas^+/+^* and *Ptch^+/-^**NRas^fl/+^* ERMS (Figure 3b top shows Western blot analyses; Figure 3b bottom shows respective quantifications). This is in contrast to our recent study, in which the oncNRAS mutation has been induced at the age of 4 weeks. In this setting, oncNRAS-expressing ERMS were more differentiated, and MYOG and TPM3 were significantly upregulated compared to the control [12]. Again, this indicates that oncNRAS-associated processes differ from each other in dependency on their occurrence during tumor development. A similar tumorigenesis-stage specificity of mutations in cancer have already been described for other genes, such as p53 [22,23], and is also in line with the “sweet spot” hypothesis of oncogenic RAS signaling in tumors [10]. Thus, our current data suggest that oncNRAS mutations accelerate tumor growth when targeting immature ERMS precursors within a specific time window (sweet spot), in which the committed ERMS precursor cells are molecularly different and permissive to the mutation.

## 3. Materials and Methods

### 3.1. Animal Experiments

Mice harboring a heterozygous *Ptch* germline mutation (*Ptch^+/-^* mice; for generation see [24]) and *Myf5^CreER^* mice [13] were on a Balb/cJ background to achieve high ERMS susceptibility [11,12,25]. Mice conditionally expressing oncNRAS (*NRAS LSL-G12D* [26], in the text named *N**Ras^fl^* mice) were on a pure C57BL/6 background. In these mice, Cre recombination removes a floxed stop cassette and thus induces the expression of oncNRAS. Cre was activated by intraperitoneal injection of 0,75 mg tamoxifen (15 mg/mL in sterile ethanol:sunflower seed oil, 1:25, Sigma-Aldrich) at postnatal days 12 and 14 (cumulative dose 1,5 mg; as described in [27]), and all injected mice were kept for further analyses.

Mice were monitored weekly for palpable tumors for 200 days or until no-go criteria (e.g., bad general condition) required termination. All sacrificed mice were examined for non-palpable tumors and were re-genotyped. Tumors were evaluated by H&E staining. The primers used for genotyping are described in Bauer et al. [12].

The study has been approved by the Lower Saxony State Office for Consumer Protection and Food Safety (file numbers 33.9-42502-04-12/0805 and 33.14.42502-04-17/2534). 

### 3.2. NRas Recombination Assays 

Recombination at the targeted *NRas^fl^* locus was analyzed by PCR on genomic DNA isolated from normal skeletal muscle and tumor tissue using the primers mNRas-WT-For AGACGCGGAGACTTGGCGAGC and mNRas-WT-Rev GCTGGATCGTCAAGGCGCTTTTCC. Upon agarose gel electrophoresis, the 487 bp and 521 bp fragments are indicative of the wild-type *NRas* and recombined *NRas^fl^* allele, respectively [12,26].

### 3.3. Western Blot and RAS Activity Assay 

Western blots were conducted according to standard methods. Primary antibodies were pAb rabbit anti-RAS, Cell Signaling Technology (#3965S, 1:1000), pAb rabbit anti-Myogenin, Invitrogen, (#PA5-110007, 1:1000), pAb rabbit anti-Tropomyosin 3, Abcam (#ab180813, 1:1000), mAb mouse anti-Desmin, Leica Biosystems (#DES-DERII-L-CE, 1:1000) and mAb mouse anti- HSC70 (B-6), Santa Cruz Biotechnology (#sc-7298, 1;10,000) and secondary antibodies pAb goat anti-rabbit IgG/HRP, Dianova Jackson Immunoresearch (#111-035-045, 1:10,000) and pAb rabbit anti-mouse IgG/HRP, Dianova Jackson Immunoresearch (#315-035-003, 1:10,000). Signals were visualized by ECL (GE Healthcare) on an Azure c300 (Azure Biosystems, Dublin, CA, USA) imaging system. Pictures were processed with Adobe Photoshop and analyzed with ImageJ. All shown Western blots are representative of at least two independent experiments.

For the RAS activity assay, which was conducted with the Active Ras Pull-Down and Detection Kit from Thermo Fisher Scientific (#16117), proteins from homogenized tissue samples were isolated in 1X Lysis/Binding/Wash buffer from the kit. RAS activity was measured according to the manufacturer’s protocol. In short, the assay uses the RAS-binding domain (RBD) of the RAS effector kinase RAF1, which specifically binds to the GTP-bound, active form of RAS proteins with high affinity. In the assay, the RAF-RBD is in the form of a GST fusion protein, which allows for a pull-down of the RAF-RBD/GTP-RAS complex with glutathione affinity beads. The amount of activated RAS (analyzed after pull-down from 500 µg protein lysate) and of total RAS (analyzed in 30 µg protein lysate) was determined by a Western blot using the RAS specific mAb mouse anti-pan-RAS antibody provided with the kit.

### 3.4. Immunohistochemistry 

For antibody staining, paraffin-embedded tumor samples were sectioned at 5 µm. The paraffin sections were stained with a rabbit monoclonal anti-Ki67 antibody (clone B56, 1:50, from Becton Dickinson). Antigen retrieval was performed in a microwave (600 W) with citrate acid buffer at pH 6.0. Antibody binding was visualized using the EnVision+ system-HRP (Dako, Santa Clara, CA, USA) and AEC chromogen. The numbers of Ki67^+^ and Ki67^neg^ nuclei were determined on 6 randomly chosen tumor areas each from 5 *Ptch^+/-^**NRas^fl/+^* and 5 *Ptch^+/-^**NRas^+/+^* mice at 200-fold magnification on an Olympus BX 60 microscope equipped with cellSens Dimension software (Olympus, Shinjuku, Japan).

### 3.5. Statistical Analyses

Statistical tests were conducted on GraphPad Prism 6 and are given in the respective figure legends. Data were considered significant when *p* < 0.05.

## Figures and Tables

**Figure 1 ijms-22-13377-f001:**
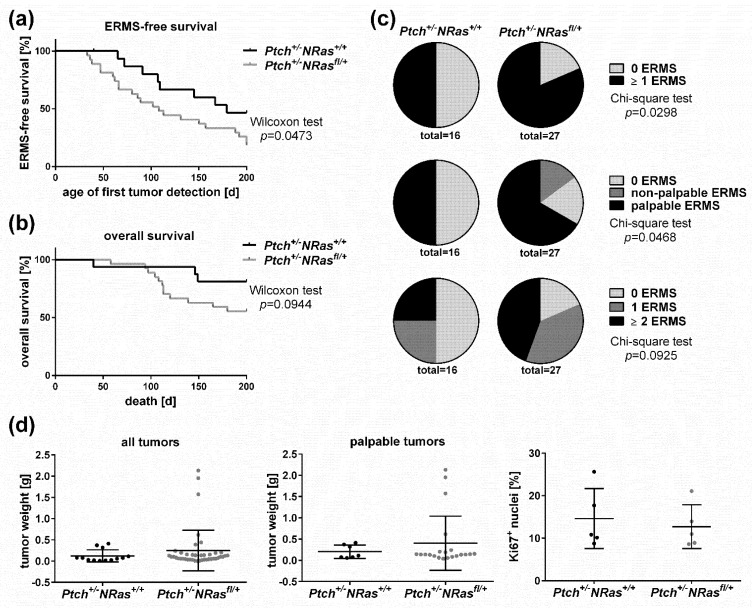
Progression of ERMS precursor lesions upon oncNRAS expression in 2-week-old mice. ERMS development in *Ptch^+/-^**NRas^fl/+^Myf5^CreER/wt^* (*Ptch^+/-^**NRas^fl/+^*) and *Ptch^+/-^**NRas^+/+^Myf5^CreER/wt^* (*Ptch^+/-^**NRas^+/+^*) mice injected with tamoxifen at 2 weeks old. (**a**) ERMS-free survival, (**b**) overall survival of the animals, (**c**) ERMS incidence of mice that ever developed an ERMS (top), incidence of palpable and non-palpable ERMS (middle) and of ERMS multiplicity (bottom) and (**d**) tumor weight of all (right) and of palpable ERMS (middle) and percentage of Ki67^+^ nuclei in ERMS tissue sections (left). Numbers of animals and tumors included in the experiments are given in Table 1. Each dot in (**d**) represents an individual tumor. For determination of the Ki67-index, all Ki67^+^ and Ki67^neg^ nuclei in 6 sections of 5 different ERMS per genotype were determined. Bars: mean ± SD. Statistical evaluation was conducted with Gehan–Breslow–Wilcoxon testing in (**a**,**c**), Chi-square testing in (**c**) and Mann–Whitney testing in (**d**).

**Figure 2 ijms-22-13377-f002:**
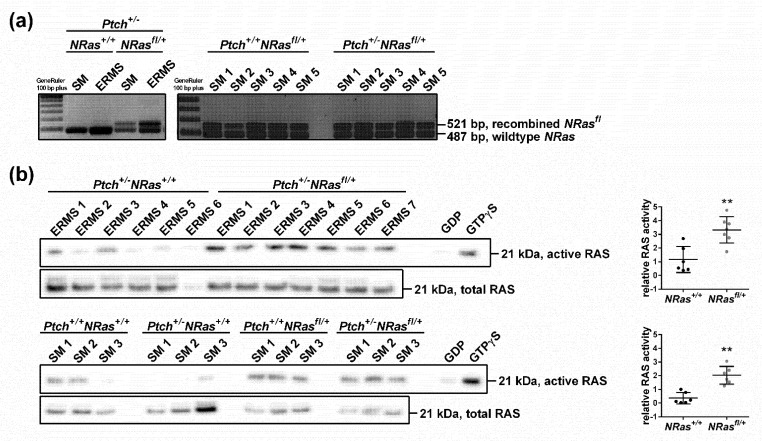
RAS activity in ERMS and normal skeletal muscle upon oncNRAS expression in 2-week-old mice. RAS activation in ERMS and skeletal muscle (SM) of *Ptch^+/-^**NRas^+/+^Myf5^CreER/wt^* (*Ptch^+/-^**NRas^+/+^*), *Ptch^+/-^**NRas^fl/+^Myf5^CreER/wt^* (*Ptch^+/-^**NRas^fl/+^*) or SM of *Ptch^+/+^**NRas^+/+^Myf5^CreER/wt^* (*Ptch^+/+^**NRas^+/+^*) and *Ptch^+/+^**NRas^fl/+^Myf5^CreER/wt^* (*Ptch^+/+^**NRas^fl/+^*) mice injected with tamoxifen at 2 weeks old. (**a**) PCR-based recombination analyses of the floxed *NRas* (*NRas^fl^*) locus on genomic DNA from SM and ERMS of *Ptch^+/-^**NRas^+/+^* and *Ptch^+/-^**NRas^fl/+^* mice (left) and from SM of *Ptch^+/+^**NRas^fl/+^* and *Ptch^+/-^**NRas^fl/+^* mice (right). (**b**) RAS pull-down assays (left) and RAS activity quantification (right, active/total RAS) of ERMS protein samples isolated from *Ptch^+/-^**NRas^fl/+^* and *Ptch^+/-^NRas^fl/+^* mice (top) and of SM protein samples isolated from *Ptch^+/+^**NRas^+/+^*, *Ptch^+/-^**NRas^+/+^, Ptch^+/-^**NRas^fl/+^* and *Ptch^+/-^NRas^fl/+^* mice (bottom). RAS pull-down of lysates incubated with GDP or the GTP analog GTPγS served as negative or positive controls, respectively. (Please note that the used RAS antibody was not isoform-specific. However, because the floxed NRAS locus is very well recombined, it is more than likely that the increase in RAS activity is due to expression of oncNRAS and not to other RAS isoforms.) Bars: mean ± SD; dots: individual tumors. Statistical evaluation was conducted by Mann–Whitney testing. ** *p* < 0.01.

**Figure 3 ijms-22-13377-f003:**
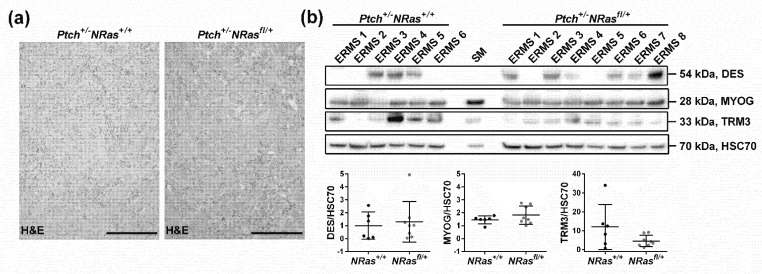
(**a**) Representative H&E staining of ERMS and (**b**) the expression of muscle differentiation markers in ERMS and normal skeletal muscle upon oncNRAS expression in 2-week-old mice. Representative Western blots analyses (top) and quantification of the results (bottom) of desmin (DES), myogenin (MYOG) and tropomyosin 3 (TRM3) expression in ERMS protein samples isolated from *Ptch^+/-^**NRas^+/+^Myf5^CreER/wt^* (*Ptch^+/-^**NRas^+/+^*) and *Ptch^+/-^**NRas^fl/+^Myf5^CreER/wt^* (*Ptch^+/-^**NRas^fl/+^*) mice, which had been injected with tamoxifen at the age of 2 weeks. HSC70 served as loading control. Bars: mean ± SD; dots: individual tumors. Statistical evaluation was conducted by Mann–Whitney testing. Scale bars in (**a**) 200 µm.

**Table 1 ijms-22-13377-t001:** Mice used for the study.

Genotype (All Mice Are *Myf5^CreER/wt^*)	Drug; Age at Application	n	^2^ Median Overall Survival (Min-Max)	Mice with ERMS (Palpable and Non-Palpable)	Mice with Palpable ERMS	Mice with ≥ 2 ERMS (Palpable and Non-Palpable)	Other Findings
** ^1^ ** ** *Ptch^+/-^NRas^fl/+^* **	no drug	26	200 days (76–206)	16 (62%)	14 (54%)	10 (38%)	^3^*cysts* (4), *medulloblastoma* (2)
** ^1^ ** ** *Ptch^+/-^NRas^fl/+^* **	5 × 1 mg Tam i.p. at P28 to P32	26	200 days (131–212)	19 (73%)	17 (65%)	8 (31%)	^3^*cysts* (6), *medulloblastoma* (1)
** *Ptch^+/+^NRas^+/+^* **	2 × 1.5 mg Tam i.p. at P12 and P14	20	201 days (193–211)	-	-	-	-
** *Ptch^+/+^NRas^fl/+^* **	2 × 1.5 mg Tam i.p. at P12 and P14	28	200 days (166–211)	-	-	-	adipositas (1), opacity of eye lens (1)
** *Ptch^+/-^NRas^+/+^* **	2 × 1.5 mg Tam i.p. at P12 and P14	16	201 (40–211)	8 (50%)	8 (50%)	4 (25%)	^3^*cysts* (5), *microsomia* (1)
** *Ptch^+/-^NRas^fl/+^* **	2 × 1.5 mg Tam i.p. at P12 and P14	27	199 (58–211)	22 (81%)	18 (67%)	12 (44%)	^3^*cysts* (9), abdominal varicosis (1), opacity of eye lens (1)

^1^ data already published in [12]. ^2^ from birth on. ^3^ intraperitoneally localized blood-filled cystic tumors.

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
