# Peer review of "Oncogenic NRAS Accelerates Rhabdomyosarcoma Formation When Occurring within a Specific Time Frame during Tumor Development in Mice"

_ijms, 2021, doi:10.3390/ijms222413377_

Round 1

Reviewer 1 Report

Overall, I think this work is properly design in order to bridge previous work of the group ("Context-dependent modulation of aggressiveness of pediatric tumors by individual oncogenic RAS isoforms") to future (determination of role of KRAS, NRAS, and HRAS during early stages of ERMS development).

However, I found few minor problems with the paper, relatively extensive problems with the language in abstract and into, and also some major methodological questions that need to be addressed.

In detail:

The English is extremely poor in the abstract and the intro.

The abstract is hard to follow due to improper English language. Same for the intro in some parts.

Lines 34-40: need to be reorganize so it flows better. Also: “fusion-negative ARMS not harboring these translocations” is basically a repetition: if you are saying fusion negative, it means there is no translocation, please fix.

Lines 134-135: satellite cells and muscle stem cells are the same thing! Please do not repeat.

Lines 133-135: I do not think your data show that muscle stem cells are induced to express oncNRAS, what is telling me that that is the case is the cited literature! Please correct the sentence accordingly. Or if you want to make the point that that is true, sort the cells and check for expression.

Lines 221-230: I am not clear how the total RAS data was obtained. Did you use the same amount of starting material as before starting the RAS activity assay? Did you use the Cell signaling 3965S antibody for detection of the Total RAS?

I am also curious to know how you can ensure that the result you are showing for RAS expression and activation are specific to NRAS, given that the antibody used and the kit are not specific to NRAS.

Also, what is really puzzling me is that despite overexpression of RAS in the floxed animals, I do not see an increased amount of total protein compared to the WT (+/+) animals. How do you explain that?

In addition, I think the discussion is pretty poor. Data are not well integrated in recent literature. I would suggest to further develop the discussion.

Reviewer 2 Report

The authors Ragab et al investigate the impact of NRAS on rhabdomyosarcoma tumor growth in early development. Mice model of ERMS were used. The results highlighted that ERMS-free survival of mice expressing oncNRAS is significantly lower compared to the control cohort. Moreover, oncNRAS increases the ERMS incidence. The authors conclude that the preliminary results suggested that oncoNRAS mutations  promote tumor growth when targeting immature ERMS precursor in a specific time.

The paper is interesting and has a good relevant to the field.

The following point should be addressed:

  1. Latest WHO should be referenced: WHO Classification of Tumours. In Soft Tissue and Bone, 5th ed.; IARC Press: Lyon, France, 2020; Volume 3, p. 368, ISBN 978-92-832-4502-5.
  2. Representative images of explanted tumors should be added.
  3. Recent findings has shed the light on the role of EMT biomarkers in Rhabdomyosarcomas tumor growth and aggressivity. In this regard the following manuscript should be referenced: Deciphering the Genomic Landscape and Pharmacological Profile of Uncommon Entities of Adult Rhabdomyosarcomas. Int J Mol Sci. 2021 Oct 26;22(21):11564. doi: 10.3390/ijms222111564. PMID: 34768995; PMCID: PMC8584142.

Minor revision are requested
